# TZAP Mutation Leads to Poor Prognosis of Patients with Breast Cancer [note 1]

**DOI:** 10.3390/medicina55110748

**Published:** 2019-11-18

**Authors:** Yu-Ran Heo, Moo-Hyun Lee, Sun-Young Kwon, Jihyoung Cho, Jae-Ho Lee

**Affiliations:** 1Department of Anatomy, School of Medicine, Keimyung University, Daegu 42601, Korea; dbfks5240@dsmc.or.kr; 2Department of Surgery, School of Medicine, Keimyung University, Dongsan Medical Center, Daegu 42601, Korea; 3Department of Pathology, School of Medicine, Keimyung University, Dongsan Medical Center, Daegu 42601, Korea

**Keywords:** TZAP, telomere, breast carcinoma, ZBTB48

## Abstract

*Background and Objectives:* ZBTB48 is a telomere-associated factor that has been renamed as telomeric zinc finger-associated protein (TZAP). It binds preferentially to long telomeres, competing with telomeric repeat factors 1 and 2. *Materials and Methods:* We analyzed the TZAP mutation in 128 breast carcinomas (BCs). In addition, its association with telomere length was investigated. *Results:* The TZAP mutation (c.1272 G > A, L424L) was found in 7.8% (10/128) of the BCs and was associated with the N0 stage. BCs with the TZAP mutation had longer telomeres than those without this mutation. Survival analysis showed that the TZAP mutation resulted in poorer overall survival. *Conclusions*: These results suggest that the TZAP mutation is a possible prognostic marker in BC.

## 1. Introduction

Breast carcinoma (BC) represents a heterogeneous group of lesions in terms of clinical, histopathological, and molecular complexity, and biological diversity [1,2]. Early diagnosis of BC would improve the prospects of survival. Consequently, an increasing number of studies have focused on biomarkers for early diagnosis and new therapeutic targets for BC [3,4,5].

Telomeres, composed of six base pair (TTAGGG) repeat sequences, are nucleoprotein complexes capping each end of the eukaryotic chromosome [6]. In normal human somatic cells, telomeres have an average length of 5 to 15 kilobases and are shortened by ~30 to 200 base pairs at every cell division [6,7]. With continuous shortening, telomeres eventually reach a critical length that triggers replicative senescence and apoptosis [8]. Therefore, telomere length (TL) must be maintained within an optimal range for long-term survival [6,7,8].

Telomere shortening is counteracted by the reverse transcriptase telomerase in stem cells and in approximately 85% of all cancers, whereas the remaining cancers maintain telomeres with an alternative lengthening mechanism [9,10]. This telomere elongation mechanism induces overly long telomeres that are cut back to normal length by rapid shortening (i.e., telomere trimming) [11]. Regulation of this process has not been identified although a recent study reports characterization of a protein that is necessary for regulating TL [12]. These authors identified the zinc finger protein ZBTB48 as a telomere-associated factor and renamed it telomeric zinc finger-associated protein (TZAP). It binds preferentially to long telomeres competing with telomeric repeat factors 1 and 2 [12]. Overexpression of TZAP causes progressive telomere shortening. TZAP localizes to chromosome 1p36, a region that is frequently rearranged or deleted in various cancers [13,14]. Genetic changes of TZAP may be associated with cancer pathogenesis; however, studies on its genetics have not been performed in any cancers. 

In the present study, we analyzed the TZAP mutation in BC. Based on previous studies [12,15], we also determined whether the TZAP mutation deregulated TL. The clinicopathological and prognostic characteristics of the TZAP mutation were investigated. 

## 2. Materials and Methods

### 2.1. Patients and DNA Extraction

One hundred and twenty-eight patients diagnosed with BC were included in the study. BCs and adjacent non-neoplastic tissues were obtained from patients undergoing surgery in Keimyung University Dongsan Medical Center (Daegu, Korea) between 2011 and 2014. Tissue samples were provided from the Keimyung Human Bio-Resource Bank, Korea. All patients were educated of the study purpose and informed consent was obtained from each study participant. The protocols were approved by the Institutional Review Board of Keimyung University Dongsan Medical Center (approval #2014-03-038-002). The pathologist confirmed the quality of BC tissues and provided them according to the study design. Patients and their distributions were classified according to the seventh and revised eighth editions of the AJCC manual. Tumor areas and adjacent normal mucosa were used for DNA extraction using an extraction kit (Absolute DNA Extraction Kit, BioSewoom, Gyeongsangnam-do, South Korea) according to the manufacturer’s instructions. DNA quantity and quality were measured using a NanoDrop 1000 (Thermo Fisher Scientific, Pittsburgh, PA, USA).

### 2.2. Telomeric Zinc Finger-Associated Protein (TZAP) Mutation Analysis

The TZAP mutation was amplified from isolated DNA using the polymerase chain reaction (PCR). PCR was performed using AmpliTaq Gold DNA polymerase (Applied Biosystems; Thermo Fisher Scientific, Waltham, MA, USA). Primer sequences are presented in Table 1. Thermocycling conditions were 40 cycles at 94 °C for 30 s, 55–57 °C for 30 s, and 72 °C for 60 s. The PCR products were separated electrophoretically on a 1.5% agarose gel and stained with ethidium bromide for 20 min to confirm the size of the bands. Direct DNA sequencing of TZAP was subsequently performed using an ABI 3730 DNA sequencer (Bionics, Seoul, South Korea).

### 2.3. Telomere Length (TL) Analysis

TL was analyzed by quantitative real-time PCR. For quantitative determination of TL relative to nuclear DNA, primers for specific amplification of the telomere (T) and nuclear DNA-encoded ß-globin (S) were selected as reported in a previous study. The primer sequences are presented in Table 1. Real-time PCR was performed on a LightCycler 480 II system (Roche Diagnostics, Mannheim, Germany). Relative TL was determined by calculating T/S values using the formula: T/S = 2-∆Ct, where ∆Ct = average Ct telomere - average Ct ß-globin. Each measurement was repeated in triplicate and then, average data was used. Five serially diluted control samples were included in each experiment. 

### 2.4. Statistical Analyses

The SPSS statistical package, version 20.0 for Windows (IBM, Armonk, NY, USA), was used for all statistical analyses. Chi-square, Fischer’s exact, and Mann-Whitney U-tests were used to analyze the relationship between variables. Survival curves, constructed using the univariate Kaplan-Meier estimators, were compared using the log-rank test. Overall survival was defined as the time between diagnosis and mortality. A two-tailed probability <0.05 was required for statistical significance.

## 3. Results

### 3.1. The TZAP Mutation in Breast Cancer

Sequences of the TZAP region were successfully analyzed in all 128 BCs. Sequencing results were analyzed by two experienced scientists (JHL and YRH) with an intraclass correlation coefficient of 0.92 (*p* < 0.001). The mutation was found in 7.8% (10/128) patients. TZAP mutations were c.1272 G > A (L424L) as silent mutation (Figure 1A). This mutation has been reported previously in hepatocellular carcinomas [16].

Clinicopathological characteristics of TZAP mutations are summarized in Table 2. The TZAP mutation was found only in the N0 stage of BCs (*p* = 0.035). This mutation was more frequently found in BC patients with longer telomeres, though it did not reach statistical significance (15.8 vs. 4.4%, *p* = 0.064). Other characteristics were not associated with TZAP mutations.

### 3.2. TL in Breast Cancer

TL was analyzed successfully using real time PCR in 128 BCs. TL was calculated in paired normal and tumor tissues. The average TL in BC was 1.10 ± 0.36 (standard deviation) when comparing the ratio of TL in tumors to that of paired normal tissues. For consideration of the association between clinicopathological characteristics and the TL and in BC, two groups, “longer” and “shorter” for TL, were assigned based on the mean TL value (Table 3) [17]. Telomere elongation was shown in 29.7% (38/128) of BCs and was significantly higher in luminal A and triple negative BCs (*p* < 0.001). The Mann-Whitney test showed that telomeres with the TZAP mutation were longer than those of wild-type TZAP (2.62 ± 1.71 vs. 0.98 ± 0.37, *p* = 0.007, Figure 1B). However, it did not have any clinicopathological value in BCs.

### 3.3. Prognostic Value of the TZAP Mutation in BC

We assessed survival to clarify the prognostic significance of the TZAP mutation in patients with BC. The median follow-up of patients for survival analysis was 46.7 months (range: 3–96 months). The results of the univariate survival analysis performed using a Kaplan–Meier curve revealed a shorter overall survival in BC patients with the TZAP mutation (52.8 vs. 87.3 months, χ^2^ = 4.37, *p* = 0.037; Figure 2A). However, TL did not have any prognostic value for patients with BC (65.2 vs. 86.7 months, χ^2^ = 0.013, *p* = 0.908; Figure 2B).

## 4. Discussion

In this study, we identified the TZAP mutation, for the first time, in BC. TZAP mutations have not yet been studied and rare mutations are only reported in the International Cancer Genome Consortium data [16]. We found that the c.1272 G > A TZAP mutation occurred frequently in patients with BC. Previously, this has been only reported by studies investigating hepatocellular carcinoma [16]; however, the same was not observed in 100 tissue samples of colorectal cancer and non-cancerous tissue (unpublished data). And we confirm that it is not single nucleotide polymorphism in many data bases. This mutation is a silent mutation, however, previous study demonstrated that silent mutation also contributes to alter transcript splicing affecting protein function as oncogenic [18]. Thus, this mutation may be BC-specific, although this needs to be confirmed.

Previous studies demonstrated that overexpression of TZAP caused progressive telomere shortening [12,15]. In agreement with these studies, our data showed that BCs with the TZAP mutation had longer telomeres than those with wild-type TZAP. It is suggested that disabling TZAP by this mutation causes dysregulation of telomere trimming [19,20]. However, the sample size with the TZAP mutation was extremely small. Therefore, additional research should be carried out to support this possible theory about the presence of longer telomeres in cancers with the TZAP mutation. We already demonstrated a positive correlation between TZAP and TERT as key regulators of telomere length in various cancers, though it was not found in BC. It also supports our hypothesis that TZAP may affect the telomere length in cancers. Additional investigations looking into how the TZAP mutation affects the molecular mechanisms of telomere crosstalk with other cellular processes are also warranted. 

The TZAP mutation and telomere length change have clinicopathological significance in BCs. Interestingly, the TZAP mutation was not found in N1 and N2 stages of BCs, suggesting that this mutation occurs in the early stage of breast carcinogenesis. However, this mutation was associated with poorer overall survival. Multivariate analysis showed that the TZAP mutation did not have independent prognostic value for BC. The Cancer Genome Atlas (TCGA) data analysis showed that lower expression of TZAP was associated with poor prognosis in BC (Figure 3) [21]. TZAP mutations may induce the loss of its expression contributing to poor prognosis. The prognosis of BC depends on many factors, such as tumor grade, estrogen and progesterone receptor expression, and the expression of human epidermal growth factor 2 [1,2,3]. Therefore, further studies with a larger number of patients, and additional data, should be performed to clarify the precise mechanism of the TZAP mutation in BC.

## 5. Conclusions

In summary, the results of our study have demonstrated, for the first time, a poorer prognosis in BC patients with the TZAP mutation. This suggests that the TZAP mutation may have an important role in the progression of BC through effects on TL. The results of the present study warrant future large-scale studies to elucidate the underlying molecular mechanisms of TZAP and to clarify this hypothesis.

## Figures and Tables

**Figure 1 medicina-55-00748-f001:**
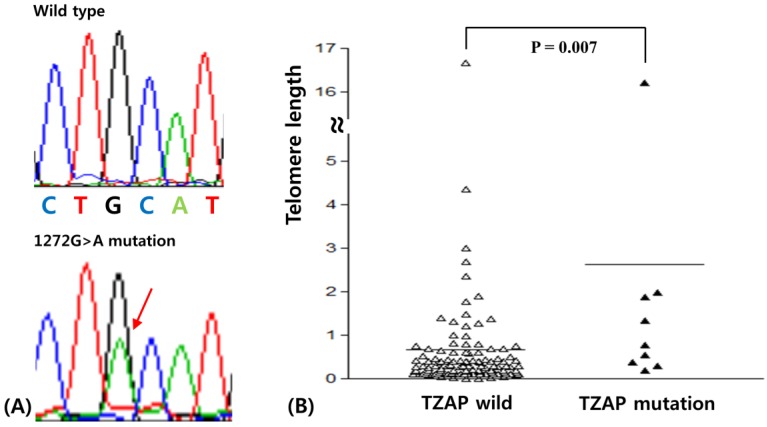
(**A**) Telomeric zinc finger-associated protein (TZAP) mutation (1272 G/A) in breast carcinomas (BCs). (**B**) Telomere length difference according to TZAP mutation status.

**Figure 2 medicina-55-00748-f002:**
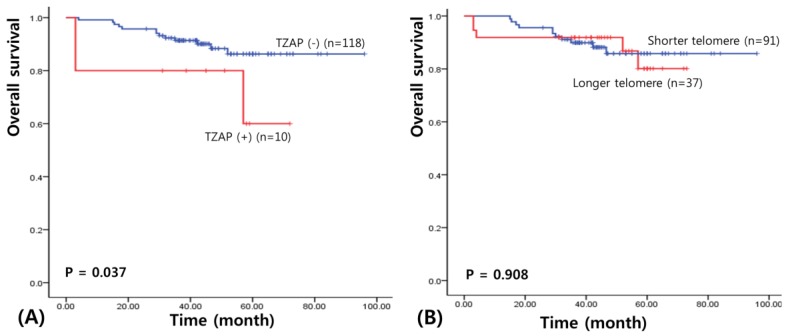
Survival analysis in BCs. (**A**) Overall survival of TZAP mutation. (**B**) Overall survival of telomere length.

**Figure 3 medicina-55-00748-f003:**
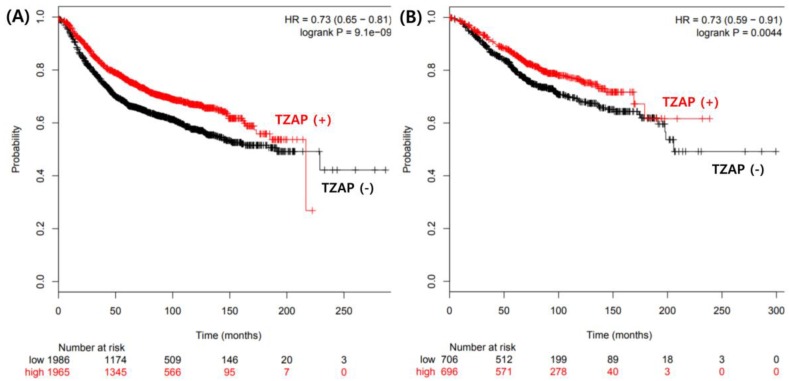
The Cancer Genome Atlas (TCGA) data analysis for prognostic value of TZAP expression. (**A**) Disease free survival. (**B**) Overall survival.

**Table 1 medicina-55-00748-t001:** Primer sequences used in this study.

	Primer Sequences	Exon
TZAP 1	F: CCAGACCTCAACAGCACAGA R: CACAGCCCACGAACCTAGTG	1
TZAP 2	F: ATCCCATTGGCCGTTCTCT R: CCGGCACAGTGAGAGGAT	2
TZAP 3	F: TAGAGGCCAACTTCCCGTTT R: CCTGGGCACAGTACCTCATT	3
TZAP 4	F: CCTGCTGATTCATTTGGTGA R: GGAATGGCAGACAGGAAAAG	4
TZAP 5	F: GGAGGTGAGGAAGTTGACCA R: CCCTTCTAAGGGGAACAAGTG	5
TZAP 6	F: GCTTGTCCCTGCACCTTAAC R: GGAGAGGGCAACACATAACC	5
TZAP 7	F: AGTCTGTCTGGGCCTGAGAA R: CCCTCCCTGTCACTTACTGC	5
TZAP 8	F: CCCTTCCCTGCTCTCACC R: AAGAGAGAACGGGCGACAC	6
TZAP 9	F: GTCACTTCCCTTGGTGATGG R: GAGGGGACCAGTGGTTTACA	7
TZAP 10	F: CTGGGTGGCACTGGAGAG R: CACGGGAACAGACTGTCAGG	8
Telomere length	F: CGGTTTGTTTGGGTTTGGGTTTGGGTTTGGGTTTGGGTT R: GGCTTGCCTTACCCTTACCCTTACCCTTACCCTTACCCT	
*β*-globin	F: TGTGCTGGCCCATCACTTTG R: ACCAGCCA-CCACTTTCTGATAGG	

F, forward; R, reverse.

**Table 2 medicina-55-00748-t002:** Clinicopathological characteristics of TZAP mutation in breast cancers.

	TZAP		
	Wild (N, %)	Mutant (N, %)	*p*
Total	118 (92.2)	10 (7.8)	
Age			0.253
<65	80 (94.1)	5 (5.9)	
≥65	38 (88.4)	5 (11.6)	
BMI			0.494
<25	44 (91.7)	4 (8.3)	
≥25	32 (86.5)	5 (13.5)	
T stage			0.706
T1	35 (89.7)	4 (10.3)	
T2	77 (93.9)	5 (6.1)	
T3	5 (83.3)	1 (16.7)	
T4	1 (100)	0 (0)	
N stage			0.035
N0	69 (87.3)	10 (12.7)	
N1	33 (100)	0 (0)	
N2	16 (100)	0 (0)	
Subtype			0.259
Luminal A	43 (87.8)	6 (12.2)	
Luminal B	38 (97.4)	1 (2.6)	
HER2-enriched	11 (100)	0 (0)	
TN	26 (89.7)	3 (10.6)	
Telomere length			0.064
Short	86 (95.6)	4 (4.4)	
Long	32 (84.2)	6 (15.8)	

BMI, body mass index; T, tumor size; N, lymph nodes; TN, triple negative.

**Table 3 medicina-55-00748-t003:** Clinicopathological characteristics of telomere length in breast cancers.

	Telomere Length
	Short (N, %)	Long (N, %)	*p*
Total	90 (70.3)	38 (29.7)	
Age			0.613
<65	61 (71.8)	24 (28.2)	
≥65	29 (67.4)	14 (32.6)	
BMI			0.127
<25	36 (75.0)	12 (25.0)	
≥25	22 (59.5)	15 (40.5)	
T stage			0.647
T1	28 (71.8)	11 (28.2)	
T2	58 (70.7)	24 (29.3)	
T3	3 (50.0)	3 (50.0)	
T4	1 (100)	0 (0)	
N stage			0.314
N0	63 (79.7)	16 (20.3)	
N1	23 (69.7)	10 (30.3)	
N2	14 (87.5)	2 (12.5)	
Subtype			<0.001
Luminal A	28 (57.1)	21 (42.9)	
Luminal B	39 (100)	0 (0)	
HER2-enriched	11 (100)	0 (0)	
TN	22 (75.9)	7 (24.1)	
TZAP mutation			0.064
Wild	86 (72.9)	32 (27.1)	
Mutant	4 (40.0)	6 (60.0)

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
