# Peer review of "TZAP Mutation Leads to Poor Prognosis of Patients with Breast Cancer †"

_medicina, 2019, doi:10.3390/medicina55110748_

Round 1
Reviewer 1 Report
In the manuscript ID: medicina-619272 – TZAP mutation leads to poor prognosis of patients with breast cancer- the authors describe the 1272G/A TZAP mutation in BC; they found this mutation to be frequently present in early stage of breast cancer, and previously, it was only reported in hepatocellular carcinoma.
These findings are of interest because the identification of specific mutation may be clinically relevant.
A few minor point should be further emphasized by the authors:
- A growing body of research has reported that TERT expression is involved in telomere shortening and cellular senescence. Is there a correlation between the TZAP mutation and/or possible reactivation of TERT expression in breast cancer?
-The TZAP mutation is present in stage N0 of breast cancer (table 2): the authors should be analyze SNP databases to increase this important information.
Author Response
Reviewer #1: the authors describe the 1272G/A TZAP mutation in BC; they found this mutation to be frequently present in early stage of breast cancer, and previously, it was only reported in hepatocellular carcinoma.
These findings are of interest because the identification of specific mutation may be clinically relevant.
A few minor point should be further emphasized by the authors:
A growing body of research has reported that TERT expression is involved in telomere shortening and cellular senescence. Is there a correlation between the TZAP mutation and/or possible reactivation of TERT expression in breast cancer?
--> Thank you for kind review. Unfortunately, we did not get RNA from these samples and there was no data for TERT expression. However, recently, I published the correlation between TZAP and TERT expression in various cancers using big data analysis. (Heo YR, Park WJ, Lee JH. Positive correlation between TZAP and TERT in most cancers: a new player in cancer diseases. Ann Transl Med. 2018;6(10):197.) However, there was no significant association in breast cancer.
-The TZAP mutation is present in stage N0 of breast cancer (table 2): the authors should be analyze SNP databases to increase this important information.
--> Thank you for kind review. I checked this mutation as SNP in NCBI SNP database. However, it is not SNP.
Reviewer 2 Report
The authors examined somatic mutation of TZAP (ZBTB48) which is associated with telomere length. Although the results are interesting, there are some concerns to be addressed.
Did the authors confirm that samples from tumor included adequate amount of carcinoma cells? If so, the methods of confirmation should be described. The authors indicated that the Mann-Whitneu test showed that telomeres wwith TZAP mutation were longer than those of wilde-type TZAP (2.62+-1.71 vs. 0.98+-0.37, P = 0.007, Fig.1B). Are these numbers mean+-SD? If so, please state in the manuscript. Furthermore, median, minimum and max value will help understanding of results. The authors stated that c. 1272G>A mutation is specific in BC (line 135). However, the authors also stated that this mutation has been previously reported in hepatocellular carcinoma. This is very confusing. c. 1272G>A mutation is silent mutation and the authors discussed about possible alteration of splicing. Are there any information about the splicing of TZAP gene?Author Response
The authors examined somatic mutation of TZAP (ZBTB48) which is associated with telomere length. Although the results are interesting, there are some concerns to be addressed.
Did the authors confirm that samples from tumor included adequate amount of carcinoma cells? If so, the methods of confirmation should be described.
--> Thank you for kind review. The Biobank provided good quality of cancer cell confirmed by pathologist (Kwon SY). Therefore, professor Kwon is included our author list. This description was added.
The authors indicated that the Mann-Whitneu test showed that telomeres wwith TZAP mutation were longer than those of wilde-type TZAP (2.62+-1.71 vs. 0.98+-0.37, P = 0.007, Fig.1B). Are these numbers mean+-SD? If so, please state in the manuscript. Furthermore, median, minimum and max value will help understanding of results.
--> Yes, they are mean +-S.D. Additional data was added.
The authors stated that c. 1272G>A mutation is specific in BC (line 135). However, the authors also stated that this mutation has been previously reported in hepatocellular carcinoma. This is very confusing.
--> This mutation was found in HCC by Big data (COSMIC) as extremely low frequency. However, its frequency was not low in breast cancers of our samples. And it was not found in other cancer tissues. So, this mutation may be specific for BC.
1272G>A mutation is silent mutation and the authors discussed about possible alteration of splicing. Are there any information about the splicing of TZAP gene?--> This mutation is silent mutation and there is no information about its function of TZAP gene. It should be studied further.
Round 2
Reviewer 2 Report
The manuscript has been well revised and acceptable for publication.